# Efficacy and Safety of a Long-Term Multidisciplinary Weight Loss Intervention under Hospitalization in Aging Patients with Obesity: An Open Label Study

**DOI:** 10.3390/nu14163416

**Published:** 2022-08-19

**Authors:** Hanan Abbas, Simone Perna, Afzal Shah, Clara Gasparri, Mariangela Rondanelli

**Affiliations:** 1Department of Biology, College of Science, Sakhir Campus, University of Bahrain, Zallaq P.O. Box 32038, Bahrain; 2Department of Chemistry, Quaid-I-Azam University, Islamabad 45320, Pakistan; 3Endocrinology and Nutrition Unit, Azienda di Servizi alla Persona “Istituto Santa Margherita”, University of Pavia, 27100 Pavia, Italy; 4Department of Public Health, Experimental and Forensic Medicine, University of Pavia, 27100 Pavia, Italy; 5IRCCS Mondino Foundation, 27100 Pavia, Italy

**Keywords:** obesity, hypocaloric diet, weight loss, multidisciplinary intervention

## Abstract

The effects of the hypocaloric diet under hospitalization on blood biochemical parameters (lipid, glycaemic, thyroid and liver profiles) were not reported in literature. This study aims to evaluate the efficacy and safety of a hypocaloric diet under hospitalisation in obese patients. A total of 151 obese subjects (49 males and 102 females, aged 69.38 ± 14.1 years, BMI 41.78 ± 7.1) were enrolled in this study. Participants were treated with an hypocaloric diet for a maximum period of 3 months. Outcomes were assessed at the beginning and at the end of the recovery period. The average duration of the hospitalisation was 47.5 days ± 1.3. The effect of the diet on all the outcomes was evaluated using the Analysis of Covariance (ANCOVA) and the predictors of weight loss were identified using linear regression. The diet induced a reduction in the anthropometric (BMI decrease of −2.713 points) and DXA body measurements in addition to serum lipids, glucose, Homeostatic Model Assessment of Insulin Resistance (HOMA-IR) and C-reactive protein (CRP) levels without affecting the muscle mass, liver and thyroid profiles. During the intervention, there was a positive shift in body composition favouring fat free mass (FFM). Lower insulin but higher serum calcium and potassium levels were predictors of weight loss.

## 1. Introduction

Obesity is a clinical condition characterised by an increase in the body’s weight relative to its height as a result of excessive fat accumulation. When energy intake exceeds energy expenditure, the excess energy is stored in fat cells, which subsequently enlarge and increase in number. Obesity is classified based on anthropometric measurements of the Body Mass Index (BMI), which is calculated by dividing weight (kg) by height squared (m^2^). For adults BMI ≥ 30 kg/m^2^ is considered obese. There are three classes of obesity: Class I BMI = 30–34.9 kg/m^2^, Class II BMI = 35–39.9 kg/m^2^ and Class III BMI = ≥ 40 kg/m^2^ [1]. BMI alone is not accurate since elevated BMI might result from high muscle mass such as the case in athletes; moreover, excess adiposity can also occur in individuals with normal weight, so the measurement of body fat is an important indicator of body composition [2]. The prevalence of obesity has increased globally over the last three decades among adults, adolescents and children. It is proposed that by 2050, 45% of the world population will be overweight and 16% obese, compared to 29% and 9% in 2010, respectively [3]. The poor health outcomes of obesity and the burden it imposes on health care systems and the economy has led to the incorporation of target 7: halt the rise of obesity by 2025 in the Sustainable Development Goals adopted by the UN general assembly [4].

This epidemic disease is associated with a number of clinical problems such as metabolic syndrome, cardiovascular diseases, diabetes mellitus, osteoarthritis, sleep apnea, infertility and cancer, as well as psychological and social problems. These outcomes result from two main mechanisms: the increased mass of fat cells or the increased levels of fatty acids and peptides secreted by the enlarged fat cells [5].

The etiology of obesity is complex, as it involves multiple factors. The susceptibility for obesity is determined by genes. Yu et al. suggested that mutations in the melanocortin-4 receptor (MC4R) regulates appetite [6]; moreover, fat distribution was reported to be linked to several SNPs [7]. In addition, other investigators identified 12 gene loci associated with BMI and waist circumference [8]. Hormones including oxyntomodulin, leptin, adiponectin, ghrelin and glucagon-like peptide can regulate appetite and food intake [9]. On the other hand, some medical conditions can lead to weight gain, such as Hypothyroidism, Cushing’s syndrome, and Polycystic Ovary Syndrome [10]. Medications such as antidepressants, beta-blockers, insulin, corticosteroids and antiepileptic drugs are also associated with obesity [11]. Behavioral factors have an important role in determining the extent of obesity, such as over-consumption of high caloric or fatty food, low physical activity [12], and skipping breakfast [13]. Sleep loss is reported to increase hunger by up-regulating the appetite-stimulating hormone, and it can also affect glucose metabolism and energy balance [14].

Given that obesity results from the interaction between multiple factors, there are a variety of different approaches reported for the treatment of this condition [15,16,17]. Lifestyle changes that include dietary, physical, and behavioral interventions are the first line treatment to manage obesity. Physical activity without diet has a very limited effect. The second line treatments include pharmacological therapy and bariatric surgery.

Caloric restriction can lead to a moderate weight loss in obese patients and evidence suggests that this approach improves renal function [18], homeostasis balance [19], myocardial function [20] and metabolic abnormalities [21,22,23]. There are no reports that evaluated the safety and efficacy of the restrictive caloric diet in obese patients under hospitalisation, taking into consideration a wide range of cardiovascular risk factors, metabolic, inflammatory and nutritional parameters.

Obesity is a chronic condition that requires continuous care, behavioural therapies and psychological support. Therefore, hospitalisation, which guarantees a multidimensional approach, appears to be a successful strategy for a weight loss program [24].

In this study, we will determine whether obese hospitalised patients (class I or II of obesity) given hypocaloric (HC) feedings with adequate protein intake can achieve a weight loss of 5–10% during 3 months of hospitalisation. Second, we will evaluate the effect of the hospitalisation in terms of different outcomes, such as lipid, glycaemic and liver profile. Finally, we will discover the possible predictor factors associated with the weight loss. The identification of the predictors of weight loss can be useful to better modulate the therapeutic path.

## 2. Materials and Methods

### 2.1. Trial Design and Setting

This is an open label study in which obese participants were treated with a restrictive hypo-caloric diet under hospitalisation for a maximum period of 3 months, with a range between 17 to 91 days, in a metabolic rehabilitation unit. The study design was approved by the ethics committee of the University of Pavia, and an individual written informed consent was obtained from each participant. Data were gathered from the end of April 2016 to the end of October 2019. All the methods were performed in accordance with the CONSORT guidelines [25]. The study was registered on ClinicalTrials.gov: NCT04622982. This study was approved by the ethics committee of the University of Pavia (approval number: 6723/22052019), and an individual written informed consent was obtained from each participant.

### 2.2. Participants

A total of 151 subjects were enrolled in this study, 49 males and 102 females. Eligible participants were aged >18 years with body mass index (BMI) ≥ 35 Kg/m^2^ without comorbidities or BMI ≥ 30 Kg/m^2^ with one or more of metabolic comorbidities (type 2 diabetes mellitus, dyslipidaemia, high blood pressure, hyperuricemia and cancer). Pregnant women and lactating women were excluded.

### 2.3. Intervention

Body weight reduction was induced by a low-energy mixed diet (55% carbohydrates, 30% lipids and 15% proteins) providing 600 kcal less than individual energy requirements based on the measured TEE. The energy content and macronutrient composition of the diets adhered to the nutritional recommendations of the American Diabetes Association [26,27].

These diets were designed to achieve weight losses of 0.5–1 kg per week; this type of diet is considered to be a low-risk intervention [28].

Individual diet plans were drawn up for each subject by the research dietitian. To optimise compliance, dietary instructions were reinforced each week by the same research dietician. Each consultation included a nutritional assessment and weighing.

Patients were administered vitamin D supplement only if they presented a value of 25-hydroxyvitamin D (25OHD) < 30 ng/mL in blood tests at the beginning. No other vitamin supplements were provided. Outcomes were assessed at the beginning (T0) and at the end of the recovery period (T1).

Rehabilitation hospitalisation lasted from 2 to 12 weeks. Patients followed different hospitalisation time in line with their clinical history, their target to achieve and clinical outcomes at weekly follow up with the medical team. This intervention targets, as the main outcome, the weight loss; as secondary targets, the hospitalisation achieves the overall improvement of all the metabolic markers, such as glycolipid profile and body composition measurements. Anthropometric parameters, such as body weight, waist and hip circumference were measured weekly.

### 2.4. Study Outcomes

#### 2.4.1. Anthropometric Measurements

Body weight, waist and hip were assessed each week during the recovery period. Body weight was measured to the nearest 0.1 kg using a precision scale; participants wore light clothing, no shoes, and a standardised method was used [29]. The waist was measured at the midpoint between the top of the hip bone (iliac crest) and lowest rib, using a standardised method.

#### 2.4.2. Body Composition

Body composition (fat free mass, fat mass and Visceral fat mass) was determined by dual-energy X-ray absorptiometry (DXA), using a Lunar Prodigy DXA (GE Medical Systems). In vivo CVs were 0.89% for whole body fat (fat mass) and 0.48% for FFM. The Skeletal Muscle Index (SMI) was taken as the sum of the fat-free soft tissue mass of arms and legs divided by height squared. Whole body and fat free mass (FFM) were divided by height squared to obtain FFM index (FFMI). FFM depletion was defined as having whole-body FFMI below the 5th centile for age- and gender-matched healthy subjects [30]. Visceral adipose tissue volume was estimated using a constant correction factor (0.94 g/cm^3^). The software automatically places a quadrilateral box, which represents the android region, outlined by the iliac crest and with a superior height equivalent to 20% of the distance from the top of the iliac crest to the base of the skull [31]. Subcutaneous abdominal fat was defined as the difference between android fat and visceral fat. The in vivo CVs were 0.89% and 0.48% for FM) and FFM, respectively [32].

#### 2.4.3. Physical Activity

The exercise programme is based on the physical activity recommendations for adults proposed by the World Health Organization [33], together with the American College of Sports Medicine’s position stand [34] on progression models in strength and aerobic training for healthy adults. Since there is limited information regarding the ideal exercise model for morbidly obese adults, we will combine strength and aerobic training (i.e., a concurrent training protocol), as previous findings in obese adults displayed important benefits when both strength and aerobic exercise are implemented in the same session [35] of 60 min of five days a week and more than 10,000 steps per day.

Physical activity was individualised and conducted every day by each subject with the help of qualified and properly-trained personal trainers also supporting motivation and self-efficacy strategies demonstrated to play a positive role in promoting physical activity.

#### 2.4.4. Behavioural and Psychodynamic Treatment

The psychodynamic approach aims at uncovering and resolving conflicts underlying the eating disorder, developing alternative coping strategies, and improving body perception and emotional expression by means of individual psychotherapy [36]. During the period of hospitalisation, according to the needs of the patient, weekly or biweekly interviews are carried out.

#### 2.4.5. Assessment of REE

Respiratory exchange measurements using indirect calorimetry (Deltatrac Monitor II MBM-200, Datex Engstrom Division, Instruments Corp. Helsinki, Finland) were used to estimate REE, adhering to the recommended measurement conditions [37].

REE was calculated from O_2_ and CO_2_ volumes—as well as from urine excretion nitrogen values—using the Weir formula, and expressed as kcal/day to obtain postprandial respiratory quotient (RQ) and substrate oxidation, and continuous gas exchange was deter-mined [38].

Carbohydrate oxidation (CO) and fat oxidation (FO) at baseline, after 30, 60, 90 and 120 min were calculated according to the Frayn equation [39].

Nitrogen excretion was estimated assuming that urinary nitrogen excretion rate was negligible. Energy expenditure from CO or FO was calculated by multiplying the individual energy expenditure data (kcal) with the individual percentage of specific substrate oxidation.

#### 2.4.6. Biochemical Analysis

Blood samples were collected at baseline and at the end of the treatment. In particular, nutritional status, lipid profile, glycaemic profile and status of inflammation were assessed.

Serum iron, lipids, uric acid, creatinine and calcium were measured by enzymatic colorimetric assay (Abbott Laboratories, Chicago, IL, USA). PCR, Transferrin, Apo A1 and Apo B were determined by immunoturbidimetry (Roche, Basel, Switzerland). ESR was measured by the Westergren method using a Diesse Analyzer, blood electrolytes by indirect ISE potentiometry (Abbott Laboratories), ionised Calcium by selective electrode potentiometry, Insulin by Electro-chemiluminescence immunoassay (ECLIA) (Roche Diagnostics). Blood glucose, aspartate aminotransferase (AST) and alanine aminotransferase (ALT) were analysed by the Enzymatic UV Assay (Abbott Laboratories) and CBC by a differential blood cell counter.

### 2.5. Statistical Analysis

The normal distribution of the study outcomes was assessed using the Shapiro–Wilk W test. The baseline characteristics of the participants were presented as means and standard deviation or numbers (%). To evaluate the effect of the hypo-caloric diet on the body weight and all the study outcomes, the Analysis of Covariance (ANCOVA) was used adjusted for age, gender and length of hospitalisation and the baseline value of each outcome. The association between change in weight (DXA) and all the outcomes was determined using linear regression (including length of hospitalisation as fixed covariate). Forward selection was applied to identify the predictors of weight loss in this group. All statistical analyses were performed using SPSS version 25.0 (SPSS Inc., Chicago, IL, USA).

## 3. Results

The baseline characteristics of the 151 subjects (32.5% males and 67.5% females) enrolled in this study is shown in Table 1A–F. The majority of the participants were middle-aged and older adults (37.1% and 55%, respectively), with the following BMI range: class I obesity (15.3%), class II (27.3%) and class III obesity (57.3%). During the treatment, the average duration of the hospitalisation was 47.5 days ± 1.3, with a range between 17 to 91 days. The biochemical tests of the blood parameters demonstrated that more than 70% of the participants had normal blood count, liver, kidney and thyroid profiles. On the other hand, vitamin D levels were mostly insufficient (53.9%) or deficient (26.5%), the Homeostatic Model Assessment of Insulin Resistance (HOMA-IR) was high in almost 60% of the participants which indicates that they were diabetic. About 42.9% of them showed low levels of high-density lipoprotein (HDL) and (55.3%) low Apo-lipoprotein A1 (ApoA1).

### 3.1. The Effect of the Hypo-Caloric Diet on the Outcomes

To study the effect of the hypo-caloric diet on the body weight and all the baseline parameters, the Analysis of Covariance (ANCOVA) was performed adjusted for age, gender and the baseline value of each outcome. Table 2 shows that the diet induced a reduction in the anthropometric and DXA body measurements with a mean body weight decrease of 5%. The loss of fat mass (−4446.88 CI95% −4874.97; −4018.8) was greater than that of fat free mass (−1772.401 CI95% −2780.5; −764.3) with 14.2% loss in the visceral adipose tissue (−339.739 CI95% −427.261; −252.216). Furthermore, levels of the liver enzymes AST, ALP and GGT were reduced −1.187 (CI95% −2.153; −0.221), −5.731 (CI95% −8.353; −3.109) and −9.115 (CI95% −12.702; −5.528) respectively. Vitamin B12 (35.407 (CI95% 11.625; 59.189)) and folic acid (4.717 (CI95% 2.939; 6.496)) increased; while parameters of diabetes and lipids were reduced except for the HDL, the effect was unfavourable because most participants had low baseline levels that were further deceased following the diet. Additionally, there was a slight reduction in haematological parameters, particularly iron and haemoglobin compared with baseline. On the other hand, thyroid hormones, SMI, ESR, uric acid, Na, K and lipase were not affected.

### 3.2. The Association between the Outcomes and the Weight Loss

The results of the linear regression demonstrated that serum levels of HOMA-IR, and potassium were significant predictors associated with the weight loss (Table 3).

Lower insulin resistance but higher calcium and potassium levels were associated with greater weight loss. Figure 1 shows that insulin resistance had the strongest association with weight loss.

## 4. Discussion

The efficacy of the hypocaloric diet in weight loss is well documented, but the effect of such diet on a whole host of blood biochemical parameters such as lipid, glycaemic, thyroid and liver profiles was not reported. In terms of the main outcome of the study, which is the weight loss, the results demonstrate that the diet induced a 5% reduction in the body weight, 9% in the fat mass and 14.2% in visceral adipose tissue. The average BMI of the study group changed from Class III at baseline to Class II at the end of the recovery period.

The beneficial effect of this diet is that it reduced body weight without affecting muscle mass, since SMI was not affected post treatment. Liver enzymes and thyroid hormones remained within the normal range post treatment. The inflammatory state was improved by lowering the CRP levels, because the excessive accumulation of macronutrients and free fatty acids in the adipose tissue, which is associated with obesity, stimulates the release of inflammatory cytokines such as TNF-α and IL-6 and decreased production of adiponectin. IL-6, in turn, triggers hepatocytes to produce and secrete CRP [40]. Reduction in CRP levels induced by weight loss post treatment with hypocaloric diets was also reported by other studies [41]. Furthermore, the diet reduced serum glucose, insulin and HOMA-IR levels, which could improve the diabetic state. This change in HOMA-IR explained almost 79% of the weight loss, suggesting that lower insulin resistance is a strong predictor of weight loss in this study group. These results are in accordance with a previous study but with a different intervention (low-carb/high protein diet) that demonstrated an improvement of insulin resistance and the metabolic syndrome in overweight diabetic patients [21], and that carbohydrate restriction decreases blood glucose. The reason for this is that higher insulin promotes the synthesis of triglycerides instead of lipolysis [42], and consequently, the accumulation of fat impairs glucose metabolism [43]. Insulin resistance is also related to the inflammation of adipose tissue, as pro-inflammatory adipokines can cause insulin resistance by disrupting insulin signalling pathways in addition to their effect on glucose metabolism [44].

Moreover, the average homocysteine levels decreased to normal range at the end of the treatment compared with being moderately abnormal at baseline; this may help in decreasing the risk of atherosclerosis. Homocysteine is one of the hepatic biomarkers of inflammation induced by the inflammatory mediators of the adipose tissue. Hyperhomocysteine can cause atherothrombosis by inducing endothelial dysfunction through oxidative stress; it is also a risk factor for diabetes. There is a bidirectional relationship between insulin resistance and homocysteine concentration [45]. This can explain our results, because reduced insulin resistance might have lowered homocysteine levels after treatment. The reduction in homocysteine might also be a result of higher vitamin B12 post treatment; since vitamin B12 plays an important role in the biochemistry and metabolism of homocysteine [46,47].

Higher calcium levels are another predictor of weight loss in this study. It is important to underline that the present study investigated only serum calcium levels and not calcium dietary intake. Other reports evaluated the intake of calcium from the diet, demonstrating that BMI is inversely related to calcium intake [48] and that dietary calcium can promote weight loss by increasing lipolysis [49]. This is the first study in the literature that demonstrates higher calcium levels as a predictor of weight loss.

In addition to HOMA-IR, the final model of the linear regression demonstrated a significant association between higher potassium levels and weight loss. As mentioned above, the present research did not evaluate dietary micronutrients’ intake, thus the association refers to serum potassium levels. These findings are in agreement with previous studies concerning dietary potassium intake, which demonstrated an association between higher potassium consumption and reduced BMI in obese patients [50,51].

Although iron, transferrin, RBC counts, haemoglobin and haematocrit percentage were slightly reduced, the magnitude of this change was not enough to cause anaemia. Damms–Machado et al. reported a decrease in serum iron concentrations in obese patients treated for a duration of three months with a standardised low-caloric formula diet, which contained 100% of micronutrients according to the Dietary Reference Intakes [52]. Iron deficiency was also observed in obese children following a very low caloric diet, whereas, other studies demonstrated that a healthy mode of weight loss is characterised by an improvement in iron concentration in obese patients [53]. Therefore, the iron and transferrin levels should be carefully monitored during hypocaloric diet treatments.

Dyslipidaemia is common in obese individuals and it is a risk factor for developing cardiovascular diseases. Lipid abnormalities include high TGs, cholesterol, LDL and Apo B, but low HDL and Apo A1 [54]. It was previously documented that a 5–10% reduction in body weight improves lipid abnormalities [55]. Our results demonstrate that the hypocaloric diet decreased the levels of all serum lipids. The unfavourable reduction—although not substantial—in HDL and Apo A1 levels was also observed in other studies following hypocaloric or very low-caloric diets [56,57]. According to a review by Rolland et al., the effect of caloric restriction on HDL is biphasic; HDL levels tend to drop initially during active weight loss then improve during weight maintenance after the intervention, this decrease was related specifically to the HDL2 sub-fraction and suggested that exercise can increase the HDL levels [58].

On the other hand, plasma albumin and pre-albumin concentrations were reduced, suggesting that the protein intake might have not been adequate. In fact, low albumin levels are indicators of malnutrition, maybe due to long term protein reduction [59,60]. It is unlikely that this reduction in albumin was due to inflammation or liver or kidney dysfunction, because the biochemical parameters related to the liver and kidney profiles of the participants remained normal after the treatment.

The novel contribution of this study is represented by the complexity of the treatment for obesity, including the choice of the setting, such as a multidisciplinary weight loss program in regimen of hospitalisation, together with the analysis of the predictors of weight loss.

The main limitation is the fact that the study design did not include a control group to validate our results. Furthermore, the study population is predominately females, middle-aged to older adults and mostly diabetic. Therefore, the results might not be generalised to obese populations different than this group. The treatment did not last for more than 3 months so our results do not determine the sustained effects of this diet on all the outcomes measured.

## 5. Conclusions

Based on our findings, we can conclude that caloric restriction is an effective and well-tolerated treatment for weight loss in this study group. Lower insulin but higher serum calcium and potassium levels were predictors of weight loss, therefore, managing the insulin resistance, calcium and potassium levels in obese patients undergoing hypocaloric diet treatment can induce better outcomes in terms of weight loss. It is recommended to monitor the iron levels and bone density during the treatment to avoid the development of anaemia and minimise bone loss. Protein intake should be increased to prevent malnutrition.

## Figures and Tables

**Figure 1 nutrients-14-03416-f001:**
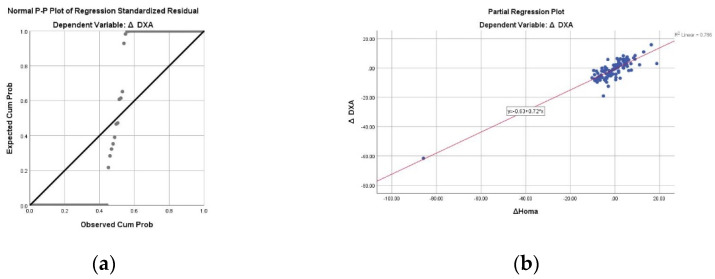
Scatter plots of the outcomes associated with the weight loss: (**a**) Normal P-P plot of standardised residual; (**b**) HOMA-IR; (**c**) Ca and (**d**) K. All data were adjusted in a stepwise model for age, gender and baseline value of each outcome.

**Table 1 nutrients-14-03416-t001:** (**A**) Baseline demographic characteristics of the study group. (**B**) Baseline anthropometric and body measurements of the study group. (**C**) Baseline nutritional and inflammatory parameters of the study group. (**D**) Baseline serum metabolites, electrolytes ions and lipids of the study group. (**E**) Baseline serum enzymes and hormones of the study group. (**F**) Baseline haematological parameters of the study group.

**(A)**		
**Characteristic**	**Minimum–Maximum**	
Age (years) (n = 151)	18–81	69.38 (14.1)
Gender (n = 151)		
Male		49 (32.5%)
Female		102 (67.5%)
Duration of hospitalisation (days) (n = 148)	17–91	47.47 (15.6)
* BMR (mean Kcal/day/kgBW) (n = 51)	30.7–70.9	1490.76 (427.3)
RQ (n = 48)	0.64–0.92	0.794 (0.09)
**(B)**		
**Characteristic**	**Minimum** **–** **Maximum**	
**Anthropometric Measurements**		
BMI (kg/m^2^) (n = 151)	30.7–70.9	41.87 (7.1)
Class I		23 (15.3%)
Class II		41 (27.3%)
Class III		86 (57.3%)
Arm Circumference (cm) (n = 99)	28–58	37.3 (4.3)
Calf Circumference (cm) (n = 99)	32–62	42.6 (5)
Waist Circumference (cm) (n = 151)	94–164	122.99 (14.1)
Hips Circumference (cm) (n = 143)	105.5–162	127.6 (12.4)
**DXA Measurements**		
FFM (g) (n = 144)	23,218.0–82,728.0	51,407.4 (10,847.9)
FM (g) (n = 144)	13,184.0–84,666.0	49,220.3 (11,468.6)
FM (%) (n = 144)	34.2–61.5	48.9 (5.7)
FFMI (n = 113)	14,749.2–28454.2	20,046.5 (2460)
FMI (n = 114)	4632.5–31,098.6	19,225.1 (4621.1)
Weight (DXA) (n = 144)	68.3–157.1	103.7 (18.9)
VAT (g) (n = 111)	960–5550	2398.7 (943.2)
SMI (kg/m^2^) (n = 142)	6.89–15.7	9.7 (1.5)
T-Score Femur (n = 96)	−2.5–2.5	−0.312 (1.3)
**(C)**		
**Parameter**	**Reference** **Minimum–Maximum**	
Folic acid (ng/mL) (n = 105)	1.3–40	8.1 (8.3)
Normal (2.7–17)		90 (85.7%)
Low		7 (6.7%)
High		8 (7.6%)
Low		16 (14.3%)
Vitamin B12 (ng/mL) (n = 108)	100–833	350.1 (143.2))
Normal (200–900)		93 (86.1%)
Low		15 (13.9%)
Transferrin (mg/dL) (n = 95)	68–441	258.8 (57.5)
Normal (170–370)		90 (94.7%)
Low		1 (1.1%)
High		4 (4.2%)
Vitamin D (ng/mL) (n = 102)	3–62.7	19.2 (12.8)
Normal (30–100)		20 (19.6%)
Insufficient (10–30)		55 (53.9%)
Deficient (<10)		27 (26.5%)
ESR (mm/h) (n = 102)	1–77	22.96 (18)
Normal Males (0–20) Females (0–30)		68 (66.7%)
High		35 (34.3%)
CRP (mg/L) (n = 121)	0.01–5.45	0.8 (1.04)
Normal (0–3)		115 (95%)
High		6 (4%)
Homocysteine (µmol/L) (m = 97)	6.8–101.2	19.5 (12.4)
Normal (<15)		30 (30.6%)
Moderate (15–30)		64 (65.3%)
Intermediate (30–100)		3 (3.1%)
High (>100)		1 (1%)
Glucose (mg/dL) (n = 142)	66–253	101.13 (28.7)
Low (<79)		13 (9.2%)
Normal (80–100)		82 (57.7%)
Pre-diabetic (101–126)		31 (21.8%)
Diabetic (>126)		16 (11.3%)
Insulin (mcIU/mL) (n = 114)	1.76–49.50	16.2 (8.9)
Normal (2.6–24.9)		95 (83.5%)
Low		1 (0.9%)
High		18 (15.7%)
HOMA-IR (mass units) (n = 112)	0.89–15.03	4.1 (2.7)
Normal (0.5–1.8)		17 (15.5%)
Early insulin resistance (1.9–2.9)		30 (25.5%)
Significant insulin resistance (>2.9)		65 (59.1%)
Pre-albumin (mg/dL) (n = 127)	7–38	23.9 (5.1)
Normal (15–36)		121 (95.3%)
Low		3 (2.3%)
High		3 (2.3%)
Albumin (g) (n = 137)	2.25–4.97	3.9 (0.38)
Normal (≥3.5)		122 (88.7%)
Low		17 (11.3%)
**(D)**		
**Parameter**	**Minimum–Maximum**	
Uric acid (mg/dL) (n = 139)	3.7–10.5	6.5 (1.5)
Normal (3–6)		87 (62.6%)
High		52 (37.4%)
Creatinine (mg/dL) (n = 139)	0.58–2.07	0.89 (0.27)
Normal Males (0.7–1.3) Females (0.6–1.1)		119 (85.6%)
Low		4 (2.9%)
High		16 (11.5%)
Total bilirubin (mg/dL) (n = 134)	0.18–2.56	0.75 (0.37)
Normal		123 (91.8%)
High		11 (8.2%)
Na (mEq/L) (n = 140)	135–144	139.6 (2)
Normal (135–145)		140 (100%)
K (mmol/L) (n = 140)	2.9–5.6	4.4 (0.43)
Normal (3.5–5.0)		125 (89.3%)
Low		2 (1.4%)
High		13 (9.3%)
Cl (mmol/L) (n = 139)	92–116	103.7 (3.5)
Normal (96–106)		110 (79.1%)
Low		2 (1.4%)
High		27 (19.4%)
Ca (mg/dL) (n = 138)	8–10.6	9.3 (0.5)
Normal (8.5–10.5)		132 (95.7%)
Low		5 (3.6%)
High		1 (0.7%)
Total Cholesterol (mg/dL) (n = 140)	63–372	185.9 (43.2)
Normal (<200)		99 (70.7%)
High (200–240)		29 (20.7%)
Very high (>240)		12 (8.6%)
HDL (mg/dL) (n = 140)	24–80	45.4 (12.1)
Normal (>60)		18 (12.9%)
Low (40–60)		60 (42.9%)
Very low (<40)		62 (44.3%)
Triglycerides (mg/dL) (n = 140)	40–378	142.5 (67.7)
Normal (<150)		94 (67.1%)
Borderline high (150–200)		25 (17.9%)
High (201–500)		21 (15%)
LDL (mg/dL) (n = 138)	25–304.8	112 (42.1)
Normal		104 (75.4%)
High		27 (19.6%)
Very high		7 (5.1%)
ApoA (mg/dL) (n = 132)	84–250	134.8 (27.3)
Normal Males (≥120) Females (≥140)		59 (44.7%)
Low		73 (55.3%)
ApoB (mg/dL) (n = 132)	26–209	102.7 (29)
Normal (<99)		67 (50.8%)
High (100–139)		52 (39.4%)
Very high (≥140)		13 (9.8%)
**(E)**		
**Parameter**	**Minimum–Maximum**	
AST (IU/L) (n = 139)	10–109	21.5 (11.5)
Normal Males (10–40) Females (9–32)		130 (93.5%)
High		9 (6.5%)
ALT (U/L) (n = 139)	7–204	28.3 (22.8)
Normal (<56)		131 (94.2%)
High		8 (5.8%)
GGT (U/L) (n = 139)	6–170	35 (30.3)
Normal (<48)		114 (82.0%)
High		25 (18%)
ALP (U/L) (n = 120)	3.5–189	66.2 (26.7)
Normal (20–140)		116 (76.8%)
Low		2 (1.3%)
High		2 (1.3%)
Lipase (U/L) (n = 110)	4–128	25.6 (17.8)
Normal (<70)		107 (97.3%)
High		3 (2.7%)
Amylase (U/L) (n = 118)	18–115	51.7 (19.7)
Normal (23–140)		115 (97.5%)
High		3 (2.5%)
TSH (µU/mL) (n = 109)	0–9.32	2.2 (1.5)
Normal (0.4–4.0)		96 (88.1%)
Low		6 (5.5%)
High		7 (6.4%)
FT3 (pmol/L) (n = 49)	1.94–3.84	2.9 (0.5)
Normal (3.5–7.8)		44 (89.8%)
Low		5(10.2%)
FT4 (pmol/L) (n = 57)	7.7–18	12.4 (1.8)
Normal (9–25)		55 (96.5%)
Low		2 (3.5%)
**(F)**		
**Parameter**	**Minimum–Maximum**	
WBC (K/µL) (n = 138)	3.6–14.2	7 (1.8)
Normal (4–11)		133 (96.4%)
Low		2 (1.4%)
High		3 (2.2%)
Lymphocytes (%) (n = 135)		33.2 (7.3)
RBC (M/µL) (n = 138)	3.2–7.04	4.7 (0. 6)
Normal Males (4.7–6.1) Females (4.2–5.4)		100 (72.5%)
Low		31 (22.5%)
High		7 (5.1%)
Hb (g/dL) (n = 138)	10.1–17.9	13.4 (1.4)
Normal Males (13.5–17.5) Females (12–15.5)		106 (76.8%)
Low		30 (21.7%)
High		2 (1.5%)
HCT (%) (n = 138)	30.8–54.8	41.1 (4.1)
Normal Males (38.3–48.6)) Females (35.5–44.9))		106 (76.8%)
Low		14 (10.2%)
High		18(13%)
MCV (fL) (n = 138)	60.8–104.7	87.2 (6.3)
Normal (80–96)		119 (86.2%)
Low		11 (8%)
High		8 (5.8%)
PLT (n = 138)	66–440	253.6 (65.3)
Normal (150–400)		129 (93.5%)
Low		6 (4.3%)
High		3 (2.2%)

* BMR: Basal Metabolic Rate, RQ: Respiratory Quotient. Data are presented as means (SD) or numbers (%). BMI: Body Mass Index; FFMI: Fat Free Mass Index; FMI: Fat Mass Index; VAT: Visceral Adipose Tissue; SMI: Skeletal Muscle Index. Data are presented as means (SD) or numbers (%). ESR: Erythrocyte Sedimentation Rate; CRP: C—reactive protein; HOMA-IR: Homeostatic Model Assessment of Insulin Resistance. Data are presented as means (SD) or numbers of patients (%). HDL: High Density Lipoprotein; LDL: Low Density Lipoprotein; ApoA1: Apolipoprotein A1; ApoB: Apolipoprotein B. Data are presented as means (SD) or numbers of patients (%). AST: Aspartate Aminotransferase; ALT: Alanine Aminotransferase; GGT: Gamma-glutamyl Transferase; ALP: Alkaline Phosphatase; TSH: Thyroid Stimulating Hormone; FT3 and FT4: Free Triiodothyronine 3 and 4. Data are presented as means (SD) or numbers of patients (%). WBC: White Blood Cells; RBC: Red Blood Cells; Hb: Haemoglobin; HCT: Haematocrit; MCV: Mean Corpuscular Volume; PLT: Platelets. Data are presented as means (SD) or numbers of patients (%).

**Table 2 nutrients-14-03416-t002:** ANCOVA analysis of the outcomes.

Outcome	Mean Difference (95%CI)
BMR (cal/day) (n = 51)	−121.4 (−188.5; −54.3)
RQ (n = 48)	0.3 (0.0; 0.0)
**Anthropometric Measurements**	
BMI (points) (n = 151)	**−2.7 (−2.9; −2.5)**
Arm Circumference (cm) (n = 99)	**−1.9 (−2.3; −1.4)**
Calf Circumference (cm) (n = 99)	**−1.2 (−1.4; −1.0)**
Waist Circumference (cm) (n = 151)	**−6.4 (−7.0; −5.9)**
Hips Circumference (cm) (n = 143)	**−4.9 (−5.5; −4.2)**
DXA Measurements	
FFM (g) (n = 144)	**−1772.4 (−2780.5; −764.3)**
FM (g) (n = 144)	**−4446.9 (−4875.0; −4018.8)**
FM (%) (n = 144)	**−2.0 (−2.4; −1.7)**
FFMI (n = 113)	**−592.4 (−1010.7; −174.2)**
FMI (n = 114)	**−1824.5( −2146.0; −1503.0)**
Weight (DXA) (n = 144)	**−5.9 (−6.4; −5.3)**
VAT (g) (n = 111)	**−339.7 (−427.3; −252.2)**
SMI (kg/m^2^) (n = 142)	−0.17 (−0.3; 0.0)
**Biochemical parameters**	
Folate (ng/mL) (n = 105)	**4.7 (2.9; 6.5)**
Iron (µg/dL) (n = 112)	**−14.9 (−18.1; −11.7)**
Vitamin B12 (ng/mL) (n = 108)	**35.4 (11.6; 59.2)**
Transferrin (mg/dL) (n = 95)	**−26.0 (−30.8; −21.1)**
Vitamin D (ng/mL) (n = 102)	**13.0 (10.5; 15.6)**
ESR (mm/hr) (n = 102)	1.4 (−0.8; 3.6)
CRP (mg/L) (n = 121)	**−0.2 (−0.4; −0.1)**
Glucose (mg/dL) (n = 142)	**−11.3 (−13.4; −9.1)**
Insulin (mcIU/mL) (n = 114)	**−2.5 (−4.2; −0.9)**
HOMA-IR (mass units) (n = 112)	**−1.1 (−1.5; −0.7)**
Uric acid (mg/dL) (n = 139)	−0.1 (−0.3; 0.1)
Creatinine (mg/dL) (n = 139)	0.1 (0.0; 0.1)
Na (mEq/L) (n = 140)	0.3 (−0.0; 0.6)
K (mmol/L) (n = 140)	−0.0 (−0.1; 0.0)
Cl (mmol/L) (n = 139)	**0.5 (0.0; 0.9)**
Ca (mg/dL) (n = 138)	**0.1 (0.0; 0.2)**
Total Cholesterol (mg/dL) (n = 140)	**−25.0 (−29.2; −20.7)**
HDL (mg/dL) (n = 140)	**−4.7 (−5.7; −3.8)**
Triglycerides (mg/dL) (n = 140)	**−22.8 (−29.3; −16.3)**
LDL (mg/dL) (n = 138)	**−12.5 (−17.0; −8.0)**
ApoA (mg/dL) (n = 132)	**−15.7 (−18.1; −13.4)**
ApoB (mg/dL) (n = 132)	**−14.2 (−17.1; −11.3)**
AST (IU/L) (n = 139)	**−1.2 (−2.2; −0.2)**
ALT (U/L) (n = 140)	−1.2 (−3.2; 0.8)
GGT (U/L) (n = 139)	**−9.1 (−12.7; −5.5)**
Pre-albumin (mg/dL) (n = 127)	**−1.8 (−2.3; −1.3)**
ALP (U/L) (n = 120)	**−5.7 (−8.4; −3.1)**
Total bilirubin (mg/dL) (n = 134)	**−0.1 (−0.2; −0.1)**
Lipase (U/L) (n = 109)	2.5 (−0.5; 5.5)
Amylase (U/L) (n = 118)	**4.7 (2.4; 6.9)**
Homocystein (µmol/L) (n = 97)	**−2.9 (−4.0; −1.9)**
TSH (µU/mL) (n = 109)	0.5 (−0.4; 1.5)
FT3 (pmol/L) (n = 48)	−0.0 (−0.1; 0.1)
FT4 (pmol/L) (n = 57)	0.4 (−0.0; 0.9)
Albumin (g) (n = 137)	**−0.1 (−0.1; −0.0)**
WBC (K/µL) (n = 138)	**−0.7 (−0.9; −0.5)**
Lymphocytes (%) (n = 135)	**2.8 (1.7; 3.8)**
RBC (M/µL) (n = 138)	**−0.1 (−0.2; −0.1)**
Hb (g/dL) (n = 138)	**−0.2 (−0.3; −0.1)**
HCT (%) (n = 138)	**−0.9 (−1.5; −0.3)**
MCV (fL) (n = 138)	**0.4 (0.1; 0.8)**
PLT (n = 138)	**−24.3 (−29.1; −19.4)**

Data have been adjusted at baseline for gender, age and for length of hospitalisation. BMR: Basal Metabolic Rate; BMI: Body Mass Index; FFMI: Fat Free Mass Index; FMI: Fat Mass Index; VAT: Visceral Adipose Tissue; SMI: Skeletal Muscle Index; ESR: Erythrocyte Sedimentation Rate; CRP: C-Reactive Protein; HOMA-IR: Homeostatic Model Assessment of Insulin Resistance; HDL: High Density Lipoprotein; LDL: Low Density Lipoprotein; ApoA1: Apolipoprotein A1; ApoB: Apolipoprotein B; AST: Aspartate Aminotransferase; ALT: Alanine Aminotransferase; GGT: Gamma-glutamyl Transferase; ALP: Alkaline Phosphatase; TSH: Thyroid Stimulating Hormone; FT3 and FT4: Free Triiodothyronine 3 and 4; WBC: White Blood Cells; RBC: Red Blood Cells; Hb: Haemoglobin; HCT: Haematocrit; MCV: Mean Corpuscular Volume; PLT: Platelets.

**Table 3 nutrients-14-03416-t003:** Linear regression with stepwise analysis of the main predictors associated with the weight loss.

Outcome	B	*p*-Value	CI_95%_
∆ HOMA-IR	1.322	<0.001	1.218; 1.426
∆ Ca	−5.858	0.001	−7.093; −4.623
∆ K	−1.499	0.017	−2.479; −0.519

Data were controlled for all covariates displayed at baseline (including length of hospitalisation).

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
