# Peer review of "Efficacy and Safety of a Long-Term Multidisciplinary Weight Loss Intervention under Hospitalization in Aging Patients with Obesity: An Open Label Study"

_nutrients, 2022, doi:10.3390/nu14163416_

Round 1
Reviewer 1 Report
This is an interesting study about the effect of weight loss diet on hospitalized patients who presented a high grade of obesity.
I have some concerns for the authors:
1. Participants. did this study open for everyone who presented obesity and was adult? In my opinion, the authors should give further details about the inclusion and exclusion criteria, for example, pregnant women or lactating women were excluded? People with other diseases such as renal or liver diseases were excluded? Please specify in participants section.
2. The reduction of 600 kcal was applied for everyone?
3. How did the authors carry out the psychological intervention?
4. Why did the authors use a regression analysis for studying the association between the outcomes and weight loss? The association should be analyzed using correlations.
5. The figure 1 c and d shows a clear outlier that maybe is "sweeping along" the result. Did the authors try to remove it?
6. Nowadays there is a high scientific evidence showing that hypocaloric diets can treat obesity with a high efectiveness, the authors should include in discussion section what is the novel contribution of this investigation.
Minor comments:
please check some typos, for example lines 122 and 199.
please modify the tables and use just one decimal value.
please modify the size of figure 1. The X axis and p value are difficult to read.
Author Response
We revised the manuscript with modifications and changes based on the reviewer’s comments.
We send you the revised manuscript together with our point-by-point response.
The changes in the text are yellow highlighted.
Thanking you in advance for your kind collaboration and suggestions.
Best regards,
The authors
REVIEWER 1
This is an interesting study about the effect of weight loss diet on hospitalized patients who presented a high grade of obesity.
ANSWER: thanks a lot for your appreciation and for having spent time on reading this paper.
I have some concerns for the authors:
ANSWER: all your concerns have been addressed into the paper.
- Participants. did this study open for everyone who presented obesity and was adult? In my opinion, the authors should give further details about the inclusion and exclusion criteria, for example, pregnant women or lactating women were excluded? People with other diseases such as renal or liver diseases were excluded? Please specify in participants section.
Answer: further details were given and better detailed into the paper. Renal or liver diseases were not excluded, because the study included obese patients with metabolic comorbidities, as better described in the text.
- The reduction of 600 kcal was applied for everyone?
Answer: yes, the reduction of 600 was applied for everyone.
- How did the authors carry out the psychological intervention?
Answer: A brief description of the psychological intervention has been added in the paragraph 2.4.5
- Why did the authors use a regression analysis for studying the association between the outcomes and weight loss? The association should be analyzed using correlations.
Answer: thanks a lot for addressing this methodological question. Our senior statistician did not apply the simple correlation because through the regression we applied the stepwise analysis in order to create a model that uncover the best predictors adjusting the model by all the variables included into the model.
With the correlation I would have been possible apply this concept.
- The figure 1 c and d shows a clear outlier that maybe is "sweeping along" the result. Did the authors try to remove it?
Answer: Yes, for figure 1c there are 2 patients with 2 data on calcium classified as outlayers. We removed them, but the results of the analysis did not change. For this reason, graphically we left these data.
- Nowadays there is a high scientific evidence showing that hypocaloric diets can treat obesity with a high effectiveness, the authors should include in discussion section what is the novel contribution of this investigation.
Answer: a new sentence has been added in the discussion section.
Minor comments:
please check some typos, for example lines 122 and 199.
please modify the tables and use just one decimal value.
Answer: Typos were corrected. Tables have been modified, using just one decimal value.
please modify the size of figure 1. The X axis and p value are difficult to read.
The layout of the journal did not allow to modify the size of figure 1. During the proofreading of the article we will provide to the assistant editor the figures in high quality so he can upload the high graphical quality figures.
Reviewer 2 Report
Thank you for the paper "Efficacy and safety of a long-term multidisciplinary weight loss intervention in ageing patients with obesity and sarcopenia: an open-label study". It is an area that requires research.
Below I have outlined some comments which I hope you will find helpful and will enhance the paper.
Introduction:
In the first paragraph of the introduction, you wrote in line 41, "measurement of body fat is required for the diagnosis of obesity'… body fat is not required for a diagnosis of obesity but can help in the interpretation of BMI.
In line 54, remove the word "many" before peptides
In lines 59 to 61, You wrote: "Several homes such as oxyntomodulin ….intake." consider changing to "Hormones including oxyntomodulin…..glucagon-like peptide to name a few, can regulate…intake." The reason is that you have written this in such a way that it implies these are the only hormones, but others are involved.
In Line 62, you should provide a reference for hypothyroidism, Cushing's syndrome, and Polycystic Ovary syndrome, causing obesity.
In lines 71-74, it reads as if lifestyle modification is the best treatment for Obesity. However, we know that greater weight loss is achieved with surgery or pharmacotherapy. Are you trying to say that lifestyle is a critical component of all treatments, including surgery and pharmacotherapy? Also, if you are trying to justify using lifestyle modifications without medications or surgery, you might discuss the cost and availability of these more extreme interventions.
In the paragraph starting line 75, you refer to several studies that have reported that caloric restriction can lead to considerable weight loss in obese patients. If you have evidence from the literature, you should cite all the "several papers" here. Additionally, I query this statement, as lifestyle typically doesn't lead to considerable weight loss but generally leads to moderate weight loss at best.
In the last paragraph of the introduction, you mention two different methods of classifying the participant's weight. Is there a reason for this? Did they have to meet both? Additional, why have you used an IBW of 22.5 instead of 25. I am assuming it's because you are investigating an Asian population. However, you then mention the Caucasian BMI cutoffs for Obesity rather than the potential Asian equivalent. (consider the Lancet WHO Consortium statement on cutoff values for the Asian population https://doi.org/10.1016/S0140-6736(03)15268-3)
I am unsure why you included the 130% as you don't use this in the results.
The introduction would benefit from including information about the following:
a) sarcopenia in obesity, as your title mentions sarcopenia, suggesting it is a vital component of the study.
b) information on why treating a patient in the hospital would be of benefit compared to treating them as outpatients or in the community.
c) information about rehabilitation hospital's role in weight management.
d) why it is essential to identify predictors of weight loss
Methods:
Throughout the document, there appears to be an excess of hyphens where you would not usually see them, e.g. com-position and cir-cumference etc. Check the whole document carefully to correct them.
It is not clear why eligibility must include having a metabolic comorbidity.
It is not clear why the intervention period varied. I assume it was because whatever the patient came into the hospital for had resolved, but perhaps include more information regarding this.
Under 2.4.1 diagnosis of Obesity
I am unsure why this section is in here. You don't mention these calculations for Obesity in the results or discussion and have only used the BMI definition where applicable.
2.4.3 body composition
Line 149, consider changing height2 to height squared.
2.5 statistical analysis
You haven't mentioned that you controlled for length of stay, but this would significantly impact outcomes. I would be surprised if it weren't a predictor.
Results:
In Table 1A: Next to BMR, do you mean Kcal/day/kgBW for the 30.7-70.9 results.
Table 1b consider changing the word (points) after BMI to Kg/m2.
I couldn't see the Bone mineral density results presented under the DXA results. Did you measure them?
In the first paragraph, Under 3.2, you mention HOmA-IR and potassium were significant predictors but have left out calcium which you then mention in the next paragraph (although the calcium plot does not look like a dose-response).
Did you control for outliers, as there appears to be 1-2 pretty influential data points that are outliers?
Discussion:
You suggest in the discussion that this study is in line with research that shows a low carbohydrate, high protein diet improves insulin resistance. However, the diet you have implemented is not a low carbohydrate or high protein diet. Its macronutrient profile is representative of a pretty standard diet.
I am not convinced, from the data presented, that calcium does predict weight loss. If there is a relationship, it could be a consequence of the weight loss rather than a predictor. More significant weight loss leads to greater bone mineral loss and thus higher calcium serum levels. Did you measure post-intervention BMD?
Research shows that to meet iron requirements without supplementation when on a hypocaloric diet, 25% of the energy needs to come from animal protein, primarily red meat.
Another limitation is that you have not controlled for length of intervention or amount of weight lost.
Conclusion:
Your conclusion appears to contain new information, e.g. regarding the need for a food frequency questionnaire to "better evaluate the association between dietary macronutrient intake and a successful weight loss" – this is unnecessary as this paper has not looked at micronutrient intakes and weight loss.
Although I think this paper could offer some interesting and valuable information to the scientific literature. I think the report would benefit from some restructuring. It is not clear from the introduction why you have done the study and what is essential about the findings. The title mentions sarcopenia but nowhere in the paper do you discuss this. There appear to be data points measured but not included, and no explanation of why.
Author Response
REVIEWER 2
Thank you for the paper "Efficacy and safety of a long-term multidisciplinary weight loss intervention in ageing patients with obesity and sarcopenia: an open-label study". It is an area that requires research.
Below I have outlined some comments which I hope you will find helpful and will enhance the paper.
Introduction:
In the first paragraph of the introduction, you wrote in line 41, "measurement of body fat is required for the diagnosis of obesity'… body fat is not required for a diagnosis of obesity but can help in the interpretation of BMI.
Answer: the sentence has been modified.
In line 54, remove the word "many" before peptides
Answer: removed.
In lines 59 to 61, You wrote: "Several homes such as oxyntomodulin.intake." consider changing to "Hormones including oxyntomodulin…..glucagon-like peptide to name a few, can regulate…intake." The reason is that you have written this in such a way that it implies these are the only hormones, but others are involved.
Answer: The sentence has been changed as suggested.
In Line 62, you should provide a reference for hypothyroidism, Cushing's syndrome, and Polycystic Ovary syndrome, causing obesity.
Answer: A reference has been added (Wilding, J.P.H. Endocrine testing in obesity. Eur. J. Endocrinol. 2020, 182, C13–C15.)
In lines 71-74, it reads as if lifestyle modification is the best treatment for Obesity. However, we know that greater weight loss is achieved with surgery or pharmacotherapy. Are you trying to say that lifestyle is a critical component of all treatments, including surgery and pharmacotherapy? Also, if you are trying to justify using lifestyle modifications without medications or surgery, you might discuss the cost and availability of these more extreme interventions.
Answer: The sentence has been modified: “Lifestyle changes that include dietary, physical, and behavioral interventions are the first line treatment to manage obesity. The second line treatments include pharmacological therapy and bariatric surgery”.
In the paragraph starting line 75, you refer to several studies that have reported that caloric restriction can lead to considerable weight loss in obese patients. If you have evidence from the literature, you should cite all the "several papers" here. Additionally, I query this statement, as lifestyle typically doesn't lead to considerable weight loss but generally leads to moderate weight loss at best.
Answer: the sentence has been modified; the citations of each study are reported in the next sentence.
In the last paragraph of the introduction, you mention two different methods of classifying the participant's weight. Is there a reason for this? Did they have to meet both? Additional, why have you used an IBW of 22.5 instead of 25. I am assuming it's because you are investigating an Asian population. However, you then mention the Caucasian BMI cutoffs for Obesity rather than the potential Asian equivalent. (consider the Lancet WHO Consortium statement on cutoff values for the Asian population https://doi.org/10.1016/S0140-6736(03)15268-3)
I am unsure why you included the 130% as you don't use this in the results.
Answer: SIMONE We have rewritten the full paragraph. We agree that 130% does not make sense.
The introduction would benefit from including information about the following:
- sarcopenia in obesity, as your title mentions sarcopenia, suggesting it is a vital component of the study.
Answer The term sarcopenia has been remove from the title.
- information on why treating a patient in the hospital would be of benefit compared to treating them as outpatients or in the community.
Answer Done
- c) information about rehabilitation hospital's role in weight management. Done
- d) why it is essential to identify predictors of weight loss. A new sentence has been added in the introduction.: “The identification of the predictors of weight loss can be useful to better modulate the therapeutic path.
Methods:
Throughout the document, there appears to be an excess of hyphens where you would not usually see them, e.g. com-position and cir-cumference etc. Check the whole document carefully to correct them.
Answer: The whole text has been grammar checked.
It is not clear why eligibility must include having a metabolic comorbidity.
Answer: Having metabolic comorbidities was not an inclusion criterion, but it wasn’t an exclusion criterion; the sentence has been clarified: “Eligible participants were aged >18 years with body mass index (BMI) ≥35 Kg/m2 without comorbidities or BMI ≥30 Kg/m2 with one or more of metabolic comorbidities (type 2 diabetes mellitus, dyslipidemia, high blood pressure, hyperuricemia, and cancer).”
It is not clear why the intervention period varied. I assume it was because whatever the patient came into the hospital for had resolved, but perhaps include more information regarding this.
Answer: Rehabilitation hospitalization lasted from two to 12 weeks in according to the results achieved by the patients. We have also detailed why patients followed different hospitalization time.
Under 2.4.1 diagnosis of Obesity
I am unsure why this section is in here. You don't mention these calculations for Obesity in the results or discussion and have only used the BMI definition where applicable.
Answer: the paragraph has been removed.
2.4.3 body composition
Line 149, consider changing height2 to height squared.
Answer: corrected.
2.5 statistical analysis
You haven't mentioned that you controlled for length of stay, but this would significantly impact outcomes. I would be surprised if it weren't a predictor.
Answer: thanks a lot for highlighting this matter. Length of hospitalization was an important covariate that we included in both models (ancova and multi linear regression).
We included an ad hoc statement into the statistical analysis section
Results:
In Table 1A: Next to BMR, do you mean Kcal/day/kgBW for the 30.7-70.9 results.
Answer: yes. We modified this issue.
Table 1b consider changing the word (points) after BMI to Kg/m2. Done.
I couldn't see the Bone mineral density results presented under the DXA results. Did you measure them? Bone mineral density is not included in these results.
In the first paragraph, Under 3.2, you mention HOmA-IR and potassium were significant predictors but have left out calcium which you then mention in the next paragraph (although the calcium plot does not look like a dose-response). Did you control for outliers, as there appears to be 1-2 pretty influential data points that are outliers?
Answer: Yes we controlled for outliers, results did not changed removing them from the model, for this reason we left them Only 2 patients showed outliers for calcium and this did not impact on the results.
Discussion
You suggest in the discussion that this study is in line with research that shows a low carbohydrate, high protein diet improves insulin resistance.
However, the diet you have implemented is not a low carbohydrate or high protein diet. Its macronutrient profile is representative of a pretty standard diet.
Answer: we modified this paragraph into the discussion.
I am not convinced, from the data presented, that calcium does predict weight loss. If there is a relationship, it could be a consequence of the weight loss rather than a predictor. More significant weight loss leads to greater bone mineral loss and thus higher calcium serum levels. Did you measure post-intervention BMD?
Answer: We did not measure the BMD post intervention because the BMD change in a period of minimum 10 months following the guidelines of Osteoporosios Society.
Research shows that to meet iron requirements without supplementation when on a hypocaloric diet, 25% of the energy needs to come from animal protein, primarily red meat.
Another limitation is that you have not controlled for length of intervention or amount of weight lost.
Answer: Yes, we controlled all pre post hospitalization adjusting for length of hospitalization. We added this explanation into the text
Conclusion:
Your conclusion appears to contain new information, e.g. regarding the need for a food frequency questionnaire to "better evaluate the association between dietary macronutrient intake and a successful weight loss" – this is unnecessary as this paper has not looked at micronutrient intakes and weight loss.
Answer: these were practical advises, but we removed in according to your comment
Round 2
Reviewer 2 Report
Thank you for your revisions they appear appropriate just two minor suggestions:
1) line 79 change the additional words "in fact" to "and"
2)in Line 128-130 you have added detail about the reason for the length of stay but it is still not clear what targets or clinical change you are discussing e.g. did they stay until they reached a weight target, and clinical change in biomarkers mentioned or because of a change in the clinical status that was outside the study?
Author Response
Thank you for the suggestions.
The changes in the text are green highlighted.
Best regards,
The authors
1) line 79 change the additional words "in fact" to "and"
Answer: the term has been changed.
2)in Line 128-130 you have added detail about the reason for the length of stay but it is still not clear what targets or clinical change you are discussing e.g. did they stay until they reached a weight target, and clinical change in biomarkers mentioned or because of a change in the clinical status that was outside the study?
A new sentence has been added to clarify the targets of intervention:
"This intervention targets, as main outcome, the weight loss; as secondary targets, the hospitalization achieves the overall improvement of all the metabolic markers, such as glycolipid profile and body composition measurements."